# Understanding the role of media in the formation of public sentiment towards the police

Rayan Succar [1,2], Salvador Ramallo[1,3], Rishita Das [1,2], Roni Barak Ventura [1,2] & Maurizio Porfiri [1,2,4 ✉]

Public sentiment towards the police is a matter of great interest in the United States, as reports on police misconduct are increasingly being published in mass and social media. Here, we test how the public's perception of the police can be majorly shaped by media reports of police brutality and local crime. We collect data on media coverage of police brutality and local crime, together with Twitter posts from 2010-2020 about the police in 18 metropolitan areas in the country. Using a range of model-free approaches building on transfer entropy analysis, we discover an association between public sentiment towards the police and media coverage of police brutality. We cautiously interpret this relationship as causal. Through this lens, the public's sentiment towards the police appears to be driven by media-projected images of police misconduct, with no statistically significant evidence for a comparable effect driven by media reports on crimes.

---

[1] Center for Urban Science and Progress, New York University, Brooklyn, NY 11201, USA. [2] Department of Mechanical and Aerospace Engineering, Tandon School of Engineering, New York University, Brooklyn, NY 11201, USA. [3] Department of Quantitative Methods, University of Murcia, Murcia 30100, Spain. [4] Department of Biomedical Engineering, Tandon School of Engineering, New York University, Brooklyn, NY 11201, USA. ✉email: mporfiri@nyu.edu

In recent years, public sentiment towards the police has been inflamed by events involving excessive use of force by law enforcement officers. Heated debates and law enforcement actions have fueled social movements in the United States (US) that play a key role in shaping public discourse, such as "Black Lives Matter"[1–3]. These movements gained significant traction following the death of George Floyd, who was murdered by police officers in Minneapolis on May 25, 2020[4,5]. Within these movements, voices to reform the police were often raised. In response to these calls, other movements that demand higher funding for the police were formed, such as "Blue Lives Matters". The social rupture regarding the police was recently reflected in a survey by the Associated Press of the National Opinion Research Center[6], which pointed to a deep divide in public view regarding the judicial trials of police officers who were allegedly responsible for individuals' injuries or deaths.

The public's response to police brutality and its impact on police-community relations particularly influence police agencies in large metropolitan areas[7], where events of police misconduct are most prevalent and police actions are subjected to greater scrutiny[8–10]. As evidence, Weitzer[11] found that high-profile incidents of police misconduct in Los Angeles and New York City had adverse effects on the public's perception of the police departments implicated in the respective incidents. Likewise, Lasley[12] showed that Rodney King's incident in Los Angeles in 1991 deeply impacted citizens' attitude towards the Los Angeles Police Department. Recently, Hagan et al.[13] proposed that legal cynicism, defined by Sampson and Bartusch as "general beliefs about the legitimacy of laws and social norms"[14], is exacerbated following events of police misconduct that lead to profound distrust in the police, especially among members of economically and racially isolated communities.

Similar to other contentious societal topics, the public's perception of police misconduct is likely not shaped by facts alone but rather by how events are portrayed by mass media and more recently by social networks – the so-called "framing effect"[15]. For example, Kapuściński and Richards[16] determined that the choice of words used in reporting the news of terrorism could modify readers' perceptions of risk, thereby creating a cognitive bias. Specifically, the authors observed that using the term "Al-Qaeda" strokes more fear in readers than the term "Domestic rebel separatist group". In controlled studies with human participants, Boudreau et al.[17] and Mullinix et al.[18] studied the response to different forms of information about police brutality, including statistics, newspaper articles, and videos. From the analysis of the administered surveys, the authors concluded that people form a stronger, negative perception towards the police after witnessing such information.

While media coverage of police misconduct is an important actor in the formation of public attitude towards the police, there exists another potential driver: media coverage of crimes[19,20]. As it is the responsibility of police departments to maintain safety and order, frequent reports on crime may negatively affect the reputation of local police. Several studies have shown that local crime per se could have a strong impact on the public's perception of the police whereby people who perceive their environment as unsafe tend to be less satisfied with the police than those who perceive their environment as safe[19–22]. However, the relationship between media coverage of local crime and the public's perception of the police is generally understudied and partially dislinked from local crimes themselves, as it is often the job of the media to sensationalize events to draw an audience[23]. News reports may exaggerate stories of criminal events, either by highlighting the blameworthiness of offenders, aggrandizing criminal situations, or by emphasizing the police's ineffectiveness in combating them[24–26]. On several occasions, research has demonstrated that the amount of media coverage of criminal events is largely uncorrelated with the actual crime rates, thereby creating the perception that crime rates continuously soar[27,28].

The important role of media in forming the public's perception towards law enforcement has not escaped the police. Police agencies widely acknowledge that police-community relationships are paramount to cultivating partnerships with citizens and creating a safer environment for them[29]. Through positive police-community relationships, the police could disseminate and receive information quickly and efficiently, recruit people and resources, and improve problem-solving processes[30–32]. In the end, the police strategically engage in symbiotic relationships with media outlets where they provide reporters with exclusive information about their activities and gain some control over their representation in the media[33,34]. Such a relationship could be used to open up the police to media inquiries and promote transparency and trust.

Whether it is driven by media coverage of police misconduct or local crime, public attitude towards the police is likely to change over time and in response to sensational events. Pinpointing the dynamic drivers of the formation of public attitude towards the police calls for overcoming the limitations of traditional survey-based research to create richer, time-resolved, multi-dimensional datasets. Such datasets could help formulate statistics-based approaches to objectively quantify how media coverage of both police brutality and local crime shape public attitude towards the police. Hillygus and Snell[35] pointed out that surveys often suffer from a lack of continuity and inconsistencies in the questionnaire design over the years. Similarly, Blossfeld et al.[36] raised practical concerns regarding recruiting participants and sampling responses.

In the past few years, social networks have emerged as a useful tool for gauging how public attitude towards various topics changes over time. It is estimated that more than 70% of the US population uses social media, whether to be informed about current events or to interact with each other[37]. Among the several social media platforms used to measure public attitude, Twitter is quickly becoming the platform of choice, offering an ultrafast "thermometer" of public attitude[38,39]. Twitter (now X) is a micro-blogging platform, where users can post visual and textual content with up to 280 characters, known as tweets. Unlike traditional survey instruments that were used in the past to assess public opinion regarding timely matters[40,41], scraping Twitter posts allows for the collection of massive datasets over long time intervals, until the very minute an event of interest may have occurred. Moreover, Twitter posts are not constrained to the language of a survey so that "respondents" can express more complex thoughts. Twitter data have been proven useful in addressing a number of problems. Without trying to be exhaustive, Buntain et al.[42] demonstrated the successful use of Twitter to gauge public attitude regarding the 2013 Boston Marathon Bombing, Lampos and Cristianini[43] used Twitter data to track the Flu pandemic, Surano et al.[44] utilized them to understand people's sentiment about the COVID lockdown in 2020, and Kraaijeveld and De Smedt[45] showed their power in forecasting cryptocurrency prices. In the context of police brutality events, Oglesby-Neal et al.[29] examined how the attitude towards the police changed after the death of Freddie Gray on April 19, 2015. Likewise, Mayes[46] used Twitter to reconstruct the public's image of the police and compare it against the image the police intended to project onto social media.

In this study, we investigated the role of the media in the formation of "public sentiment" towards the police, measured from Twitter posts. Within the context of natural language processing, sentiment is understood as a classification of a piece of text in terms of the positivity and polarity of the emotions it

reflects[47]. Thus, we operationally defined public sentiment as a collective measure of the positivity and polarity of the emotions of the public, measured by the sentiment of Twitter posts. In this vein, public sentiment serves as an approximation of the aggregate public mood towards a topic[48]. The concept of public sentiment is related but not equivalent to that of public opinion, which, according to Britannica[49], refers to the aggregate views, attitudes, and beliefs of the public toward a particular topic. In criminal justice, public opinion may entail emotional, behavioral, and cognitive aspects[50], whereas public sentiment does not necessarily reflect judgment and evaluation.

With these definitions in mind, we assembled an original dataset, consisting of 10 years of daily-resolved Twitter posts containing the public's sentiment towards the police, media coverage of police brutality, and media coverage of local crime in 18 metropolitan areas in the US, for a total of more than two and a half million tweets and nearly two hundred thousand articles. Based on the technical literature[11,20,21,29], we formulated the following hypotheses (Fig. 1a):

H1-Public discourse about the police on Twitter will be influenced by media reports of local crime. Specifically, we expected that media coverage of local crime will influence both positive (H1a) and negative (H1b) public sentiment as perceived from Twitter posts: public sentiment towards the police will become more favorable when media coverage of local crime decreases, and conversely, public sentiment towards the police will become unfavorable when media coverage of local crime increases.

H2-Public discourse about the police on Twitter will be influenced by exposure to instances of police misconduct everywhere in the US through the media. Specifically, we anticipated that Twitter users will express distrust in the police as news about police brutality breaks, so that negative sentiment will proliferate in response to increased media coverage of police brutality.

To overcome the known limitations of linear correlation analyses in the discovery of causal associations[51–53], we relied on an information-theoretic approach for hypothesis testing[54,55]. Specifically, we inferred a causal association from a source variable to a target variable in a Wiener-Granger sense, through the reduction of uncertainty in the prediction of the future of the target variable from its history due to additional knowledge about the history of the source. Within an information-theoretic approach, the uncertainty encoded in a variable is measured in terms of its entropy and causal associations are determined through transfer entropy[54]. An association that is discovered using transfer entropy does not mean that manipulation of the source variable will inevitably lead to a change in the target variable. Instead, such a link implies that the source variable helps in the prediction of the target variable, without ruling out the interference of unobserved variables[56]. Transfer entropy has been used to infer causality between two or more stochastic dynamical systems and the direction of information flow in several applications, ranging from financial markets[57] to the study of climate change[58,59].

We chose to focus our study on the 18 most populated metropolitan areas in the US (Fig. 1b), based on the following rationale. As one would expect, reliable inference of associations relies on large amounts of data[60], which can be accessed if working with metropolitan areas. Not only are rates of police-related fatalities higher in more densely populated areas[8,9], but also the attention of media to police brutality and protests about police violence gain more resonance when they take place in large cities[61]. Working with an original dataset, we put forward a mathematically backed, statistical framework that begets three main advantages with respect to the current state of knowledge. First, in contrast to the standard practice that relies on descriptive surveys of public opinion[11,21], our analysis is performed on rich, multi-dimensional time series, whose coupled evolution is indicative of their interdependencies. Second, our approach allows for investigating the influence of media coverage of local crime on the public's perception of the police across many US metropolitan areas. Third, the information-theoretic tools we employed to understand the driving factors of public sentiment towards the police overcome classical correlational analyses[62].

Since predictions from an information-theoretic analysis come with assumptions on the dynamics of the variables at hand (such as stochasticity and separability) and are prone to the presence of unobserved (latent) variables, we validated our claims through three independent analyses. First, we performed statistical analysis with highly resolved data collected in the wake of George Floyd's murder, a unique event in terms of impact and media coverage[5]. Specifically, we collected data from the most followed Twitter accounts of newspapers that were active during that period to unveil the chain of posting events between the accounts of newspapers and general users, adapting methodologies that are often used in the study of climate networks and extreme events[63,64]. Second, we applied an alternative methodology borrowed from the study of causal structures in ecology[52], which does not employ a Wiener-Granger notion of causality like the one that underlies information-theoretic schemes[54,55]. Third, we employed another causal discovery approach developed by Gerhardus and Runge[65] based on the conditional independence[66] framework, to account for the possibility of the presence of unobserved variables.

## Methods

**Ethical standards.** This study was not pre-registered. It was administratively reviewed by New York University (NYU)'s Institutional Review Board (IRB), determining that it does not meet the criteria for NYU's engagement in research involving human participants as defined by 45CFR46.102. This determination was made because the study does not engage with human participants (that is, obtaining information about living humans through interaction or intervention or obtaining identifiable private information). The data collected in May 2022 for this study does not contain information that identifies the individuals who posted materials on Twitter, such as their name, age, gender, sex, race, or ethnicity.

**Data collection and sentiment analysis.** We collected data to generate time series for four variables between October 1, 2010 and December 31, 2020: media coverage of police brutality, media coverage of local crime, positive Twitter posts about the police, and negative Twitter posts about the police.

Data on media coverage of police brutality and local crime were collected using the ProQuest search engine. While news can be consumed through multiple media sources, including television, radio, and the internet, only print media can be collected in a reliable manner[34]. Databases that systematically record data on the content of news in media other than newspapers do not exist and scraping such large-scale information from the internet is not feasible. In the present study, media coverage of police brutality constitutes a "global variable", as police brutality events have a reach that extends beyond city and state boundaries, spanning the entire US. That is, while local crimes are not typically of interest to people living elsewhere, incidents of police violence and their aftermath draw public attention at a national level. This effect was previously demonstrated by Weitzer[11], who examined the cases of

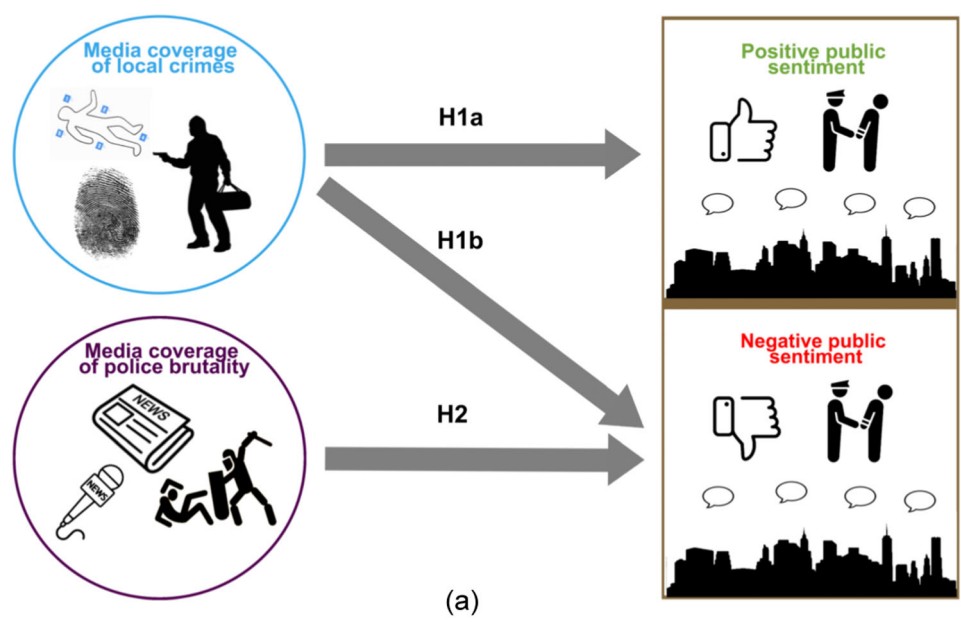

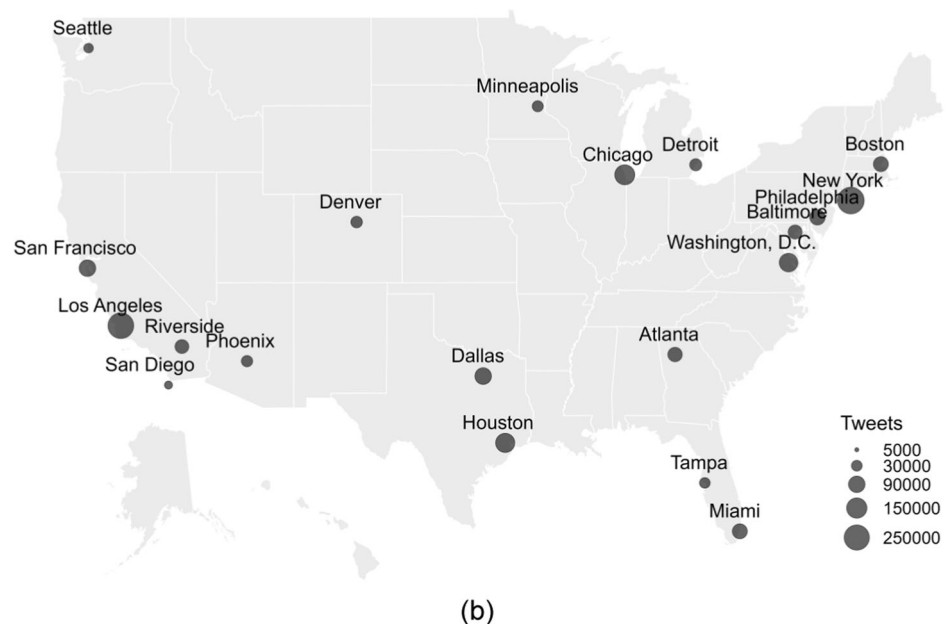

**Fig. 1 Illustration of the hypotheses and metropolitan areas under investigation. a** H1a and H1b refer to the influence of media coverage of local crime on public sentiment towards the police. H2 suggests that media coverage of police brutality, occurring anywhere in the US, drives negative public sentiment. **b** Each metropolitan area is represented with a circle, whose radius reflects the number of positive and negative tweets collected; areas were selected based on population size. Photo: 4zevar, Andrey-Kuzmin, ART.ICON, Black Creator 24, imagewriter, Oasis World, StockAppeal, sumberarto, Tatiana Garanina/Shutterstock.

Abner Louima (sexually assaulted by a police officer in 1997) and Amadou Diallo (shot by a police officer who mistakenly thought he was pulling a gun out of his pocket in 1999), and by Fridkin et al.[67], who studied the arrest of Ersula Ore (arrested for alleged assault after a confrontation with a campus police officer in 2014). Therefore, media coverage of police brutality was acquired by searching on ProQuest for the words "police" and "brutality" appearing together in sequence in the text or the headline of printed articles, published in the 20 most-circulated newspapers in the country[68]. We recorded the number of articles every day from October 1, 2010, to December 31, 2020. Since printed

articles capture events that took place the day before, the time series was shifted backward by one day.

For media coverage of local crime, we focused our search on local newspapers. We collated data for the top 20 most-populated metropolitan areas in the US[69]. For each metropolitan area, we searched for the word "crime," appearing together with the name of any of the cities with a population of more than 100,000 residents that form the metropolitan area under scrutiny. For example, for New York City metropolitan area, we searched for the word "crime" together with "New York City", "Yonkers", "Woodbridge", "New Haven", "Stamford", "Smithtown",

"Paterson", "North Hempstead", "Newark", "Huntington", "Hempstead", "Elizabeth", "Waterbury", "Oyster Bay", "Edison", "Brookhaven", "Jersey City", "Islip", "Bridgeport", or "Babylon." Similar to the search on media coverage of police brutality, we searched for printed articles, published in the 20 largest newspapers by circulation in the corresponding state available on ProQuest[70]. For example, for the metropolitan area of New York City, we searched in the following newspapers: Philadelphia Inquirer, Pittsburgh Post-Gazette, New York Times, New York Daily News, Newsday, Rochester Democrat and Chronicle, Asbury Park Press, Press of Atlantic City, Hartford Courant, Buffalo News, Pittsburgh Tribune-Review, and Home News Tribune. The daily count of news articles that met these criteria between October 1, 2010 and December 31, 2020 was registered to generate a time series for each metropolitan area; see Supplementary Note 1 for further details on the ProQuest search, including detailed queries of all the metropolitan areas.

Next, we collected data on the public's sentiment regarding the police from Twitter. Specifically, using the Python Library snscrape[71], which relies on Twitter's dedicated application programming interface (API), we scraped geo-located tweets for all the 20 metropolitan areas. The keywords used in the search included "police," "cop," and the abbreviation of the police department of the main city of the area (for example, "NYPD" for New York Police Department of the main city in the New York City metropolitan area). To identify the confined geographical boundaries of each metropolitan region, we approximated its area with circles of different radii. Tweet scraping was performed for each of these circular regions covering the complete metropolitan area. The search was conducted in May and June of 2022 and in compliance with Twitter's terms of service at the time.

Given the time series of tweets in each metropolitan area, we performed aspect-based sentiment analysis to disentangle positive and negative sentiment from tweets. While other methods for sentiment analysis do not refer to any specific word (target), aspect-based sentiment analysis classifies the sentence with respect to a target based on the rest of the sentence (context). To this end, we made use of the DeBERTa model by Yang et al.[72] through the Pytorch package, a powerful state-of-the-art pre-trained language model with exemplary performance in context modeling.

In addition to people's sentiment towards the police, we used Twitter to collect data on media coverage of police brutality with a minute-by-minute resolution. Specifically, we scraped for Twitter posts containing the term "police brutality" from Twitter accounts of the top ten most-followed newspapers in the country. Searching for tweets between October 1, 2010 and December 31, 2020 yielded a sparse time series, hence, we focused on the period around the murder of George Floyd, during which Twitter activity on the topic was intense. To avoid ceiling effects in the analysis, we began the collection five days after the murder and recorded 14 consecutive days from 00:00 on May 29, 2020 until 23:59 on June 13, 2020. This highly resolved time series was examined in conjunction with the time series of negative tweets about the police during the same two-week time window.

Two metropolitan areas (Seattle and San Diego) were excluded from the analysis due to limited data. San Diego had too few observations of media coverage of local crime (less than 0.8 per day), while Seattle's data on media coverage of local crime was concentrated in a small time window (more than one third of the total values occurred within a couple of days), thereby challenging the processes of symbolization and detrending that are needed for the implementation of the methods.

**Seasonal adjustment and detrending.** One prerequisite to transfer entropy analysis is the stationarity of the processes under

inspection. Therefore, all our daily time series were detrended and seasonally adjusted. An additive decomposition of the series using Loess regression[73] was performed with the *statsmodels* package in Python[74], specifying weekly, monthly, and yearly seasons. The resulting trend and seasonal components were subtracted from the original time series.

**Transfer entropy analysis.** We quantified causal influence within an information-theoretic approach, based on transfer entropy. Within this framework, influence is measured as the enhanced ability to predict the future of a variable from its present due to the knowledge of the present of another variable. Entropy of a random variable $X$ is its average information content (or uncertainty), defined as

$$H(X) = -\sum_{x \in \chi} P(X = x) \log P(X = x), \tag{1}$$

where $\chi$ is the sample space of the variable $X$, $x$ is any possible realization of $X$, and $P$ refers to the probability of an event $x$. When entropy is measured in bits, the base of the logarithm is set to be 2. This quantity can not be negative by construction, and zero entropy refers to a deterministic variable.

For two processes $X$ and $Y$, the joint entropy is given by

$$H(X, Y) = -\sum_{x \in \chi, y \in \gamma} P(X = x, Y = y) \log P(X = x, Y = y), \tag{2}$$

which can be understood as the global uncertainty of both variables. Similarly, the conditional entropy can be defined as

$$H(X|Y) = -\sum_{x \in \chi, y \in \gamma} P(X = x, Y = y) \log P(X = x|Y = y), \tag{3}$$

which measures the amount of information in variable $X$ given knowledge of $Y$.

Assuming $X$ and $Y$ are a pair of discrete-time stationary stochastic processes, transfer entropy from $Y$ (source) to $X$ (target) is defined as

$$TE_{Y \to X} = H(X_{t+1}|X_t) - H(X_{t+1}|X_t, Y_t). \tag{4}$$

In this formulation, if knowledge of $Y_t$ does not improve the prediction of $X_{t+1}$, then both entropy terms will be identical and transfer entropy from $Y$ to $X$ will be zero. Alternatively, if $Y$ encodes information that reduces the amount of entropy $X$ and helps predict it, transfer entropy will have a positive value.

In the multivariate case where more than two variables are considered, there might be indirect coupling between processes that can lead to detection of spurious interactions between two variables[55]. In such cases, conditional transfer entropy can be used to mitigate this issue. Let $Z_i$, for $i = 1, \ldots, m$, be other $m$ stationary processes in addition to $X$ and $Y$. Then, conditional transfer entropy from $Y$ to $X$, with $Z_i$ conditional processes is defined as

$$\begin{aligned} TE_{Y \to X|Z_1, \ldots, Z_m} = {} & H(X_{t+1}|X_t, Z_{1,t}, \ldots, Z_{m,t}) \\ & - H(X_{t+1}|X_t, Y_t, Z_{1,t}, \ldots, Z_{m,t}). \end{aligned} \tag{5}$$

Through this formulation, we control for the possibility of a process $Z_i$ acting as a common driver of $X$ and $Y$, and ensure that we do not incorrectly infer an interaction between $X$ and $Y$ due to presence of a common driver. Moreover, conditional transfer entropy also accounts for spurious information flow by a possible cascade effect where $X$ and $Y$ are indirectly coupled through $Z_i$.

From a practical point of view, there is seldom knowledge of the exact probability distributions for computation of entropy, as they are estimated from a finite time series sample. Therefore, we followed the standard practice of symbolizing our time series[75]. The symbolization was accomplished by binning the values of the time series into three equally sized quantiles, representing high,

medium and low values of the time series. The length of the time series poses a constraint on both the maximum number of processes that can be examined at once and the number of symbols used[76]. In fact, there is a trade-off between the number of symbols used and the number of processes to include, since for $s$ symbols we need to estimate $s^{(m+3)}$ conditional probability values, where $m$ is the number of confounding variables. Since the time series of media coverage contained the number of articles in daily print editions, it was shifted backwards by one day. Therefore, at a given time step $t$, a tuple of the four time series would represent the amount ("Low," "Medium," or "High") of positive tweets, and negative tweets, as well as the amount of media coverage of police brutality and crimes from printed outlets the following day.

Once the time series were symbolized, conditional transfer entropy could be estimated from Eq. (5). To determine whether the computed conditional transfer entropy value is significantly different from zero, we adopted the approach used by Runge et al.[77] and Porfiri et al.[78] Specifically, a surrogate distribution was created by shuffling the source variable while maintaining the structure of the dynamics of the target and conditioning variables. We first grouped the target variable and the conditioning variables together for every possible joint realization, and then we shuffled the symbols of the source within these groups. We computed the conditional transfer entropy for the shuffled time series, and repeated this procedure 20, 000 times to create a surrogate distribution. This distribution would represent values of transfer entropy obtained by chance. Finally, the $p$-value of the transfer entropy of the observed time series is given as the percentile of this estimated value within the ordered surrogate distribution. A transfer entropy value will be considered significantly greater than chance if it is larger than the 95th percentile of the surrogate distribution.

We computed three values of conditional transfer entropy, corresponding to the hypothesized associations under H1a, H1b, and H2. Specifically, to assess the role of media coverage of local crime (MLC) on negative tweets (NT) and positive tweets (PT) about the police posited by H1a and H1b, we computed transfer entropy from MLC to NT and from MLC to PT, conditioned on media coverage of police brutality (MPB): $TE_{\mathrm{MLC}\rightarrow\mathrm{NT}|\mathrm{MPB}}$ and $TE_{\mathrm{MLC}\rightarrow\mathrm{PT}|\mathrm{MPB}}$, respectively. To study the effect of MPB on NT underlying H2, we calculated $TE_{\mathrm{MPB}\rightarrow\mathrm{NT}|\mathrm{MLC}}$. The rationale for conditioning transfer entropy computations on a third variable was to mitigate common driver and cascading effects[55,79]; for example, by using $TE_{\mathrm{MLC}\rightarrow\mathrm{NT}|\mathrm{MPB}}$ we control for the possibility that MPB could simultaneously drive MLC and NT (common driver) or that MLC influences MPB, which, itself, influences NT (cascade).

Transfer entropy alone does not offer insight into the type of association between two variables; that is, transfer entropy does not indicate whether the association between two variables is positive (both variables are increasing or decreasing together) or negative (one variable increases while the other decreases). As such, another analysis is required for a complete hypothesis testing. Specifically, for the significant links, we computed the partial correlation between the past of the source variable and the present of the target variable, controlling for the past of the target variable and the past of the conditioning variable. To compute partial correlation, we used the Python *pingouin* package[80], which implements the method developed by Kim[81]. All correlation analyses were performed with the non-parametric Spearman correlation[82].

**Analysis through convergent cross mapping**. To offer independent backing to the transfer entropy analysis, we performed

an additional causal analysis using a dynamical systems method called convergent cross mapping (CCM)[52], which is not based on Wiener-Granger causality. CCM is a causal discovery framework developed for coupled dynamical systems. It is one of the most widely used causal inference techniques alongside Wiener-Granger causality, whereby it can be used for the detection of coupling in non-linear, non-stationary time series[83], as well as in very short time series[84]. The method has been demonstrated in animal behavior[85], neuroscience[86], ecology[87], and earth and climate science[88,89].

CCM is built on Takens' theorem for state-space reconstruction[90], which states that a time delay embedding of a time series of a dynamical system is sufficient to reconstruct a diffeomorphic attractor of the original one, also known as a shadow manifold. It proposes that the successful reconstruction of time series $X$ from a manifold of the time-delayed embedding of another time series $Y$ implies coupling between $X$ and $Y$. More precisely, if $Y$ can be used to reconstruct $X$ and not vice versa, we conclude that $X$ is driving $Y$. To reconstruct one time series from a manifold, Sugihara et al.[52] proposed the use of the simplex projection method[91], which is a linear weighted average of the nearest neighbors. Given two time series, $x_t$ and $y_t$, the manifold $M_y(t)$ of the time delay embedding of delay $\tau$ and dimension $E$ of $y_t$ is $M_y(t) = [y_t, y_{t-\tau}, y_{t-2\tau}, \ldots, y_{t-(E-1)\tau}]$ for $t = 0, \ldots, N$ where $N$ is the length of the time series. The $n$ nearest neighbors of $M_y(t)$, in the $E$-dimensional space are $T_1(t), T_2(t), \ldots, T_n(t)$, from nearest to farthest. Using $M_y(t)$, the reconstructed time series

$$\hat{x}_t | M_y(t) = \sum_{i=1}^{i=n} w_i x_{T_i(t)}, \qquad (6)$$

where $w_i$ is a weighting based on the distance between $M_y(t)$ and its $i$-th nearest neighbor $M_y(T_i(t))$,

$$w_i = \frac{\exp\left[\frac{-d[M_y(t),M_y(T_i(t))]}{d[M_y(t),M_y(T_1(t))]}\right]}{\sum_{j=1}^{j=n} \exp\left[\frac{-d[M_y(t),M_y(T_j(t))]}{d[M_y(t),M_y(T_1(t))]}\right]},$$

with $d[\,\cdot\,]$ being the Euclidean distance.

The accuracy of the reconstruction is tested by inspecting the correlation between the reconstructed time series and the original time series. While any measure of correlation can be used, we chose to apply the Pearson correlation coefficient[92]. As explained by Sugihara et al.[52], the larger the library length (that is, the number of data used for reconstruction), the denser the manifold becomes, thereby improving the correspondence of such nearest neighbors on the two manifolds. The three parameters of CCM are the embedding dimensions $E$ and $\tau$, and the number of the nearest neighbors $n$. According to Sugihara et al.[91], the minimum number of nearest neighbors that can be used is $E + 1$, which is also the one commonly used[84]. The embedding dimensions should be picked such that the attractor used in the reconstruction is best unfolded, which is accomplished with a simplex projection[84]. Herein, CCM analysis was performed using a Python code by Delforge et al.[93]

**Detection of latent variables**. Transfer entropy cannot dismiss the presence of confounding variables[56] as it would require conditioning on all sources in the set of causal information contributors[94], a virtually unfeasible task. To address the possibility of latent variables that could confound potential cause-and-effect relationship between MPB and NT, we employed the Latent Peter-Clark Momentary Conditional Independence (LPCMCI) framework[65]. LPCMCI is a state-of-the-art causal discovery algorithm for large temporal datasets based on the conditional independence framework[66]. The algorithm begins with a complete causal graph; through an iterative process of conditional

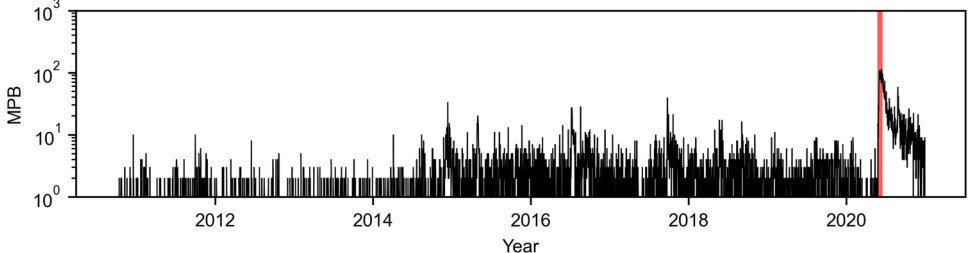

**Fig. 2 Time series of media coverage of police brutality.** Daily number of news articles mentioning "police brutality" in the 20 most-circulated newspapers in the US. A peak of 113 articles was recorded on June 13, 2020, 19 days after the death of George Floyd; activity in the wake of George Floyd's murder is highlighted in red.

independence tests that entails the removal and orientation of causal links, the algorithm removes non-significant edges until convergence. Importantly, for any pair of variables in the causal graph, the algorithm classifies the nature of their relationship and detects whether there is a latent variable that is driving both of them at the same time[65,95]. We used the publicly available Python library *Tigramite* for the detection of latent variables in all the 18 metropolitan areas using MPB, NT, and MLC. Consistent with the rest of the study, the significance level was set to $\alpha = 0.05$. The Pearson partial correlation was used as the conditional independence test, similar to CCM.

**High-resolution analysis in the wake of George Floyd's murder.** For each metropolitan area and each Twitter post by newspapers on police brutality, we recorded NT about the police by users in a time window centered about the newspaper post. Each tweet from the media was assigned two numbers: the average number of users' tweets per minute before the post and the average number of users' tweets per minute after the post. As we sought to quantify the potential increase in the NT after a media tweet about police brutality, we performed a one-tailed Wilcoxon signed-rank test comparing the average NT per minute before and after each of the 205 instances of media tweets on police brutality. The null hypothesis of the test was that the two distributions share the same mean, and the alternative hypothesis was that the mean of the distribution of the average NT after media posts is greater than the mean of the other distribution. The same procedure was performed with several time intervals whose semi-width was systematically varied from 30 to 720 minutes, with 30-minute increments.

**Reporting summary.** Further information on research design is available in the Nature Portfolio Reporting Summary linked to this article.

## Results

**Data collection.** Time series were generated for the four variables (MPB, MLC, PT, and NT), each consisting of 3745 measurements. MPB was a single daily time series for the entire country that aggregates the number of newspaper articles from 17 out of the 20 most-circulated newspapers in the country (Fig. 2). MLC assembled 18 daily time series of the number of newspaper reports on local crime, one for each metropolitan area. For the sake of illustration, we present time series for the metropolitan area of New York City (Fig. 3a) and aggregated statistics for all the metropolitan areas (Fig. 4); all other times series are included in Supplementary Note 2 (Supplementary Figs. 1–17). For PT and NT, we report the daily sampled time series of the New York City metropolitan area as an example (Fig. 3b, c, respectively). For other metropolitan areas, we present aggregated statistics (Fig. 4); the complete set of times series is included in Supplementary

Note 2 (Supplementary Figs. 1–17). Finally, Fig. 5 illustrates the highly resolved time series of MPB after George Floyd's murder.

**Transfer entropy analysis.** Results of transfer entropy analysis for all metropolitan areas (Table 1) offer strong support in favor of H2 (the null hypothesis of independence was always rejected). On the contrary, partial support in favor of H1a was gathered for the metropolitan areas of Dallas and Washington, D.C. (the null hypothesis of independence was not rejected for all the other 16 metropolitan areas), while hypothesis H1b was only supported for the metropolitan area of Dallas, the null hypothesis of independence was rejected for all the other 17 metropolitan areas. For each of the significant associations, we computed the partial correlation and found a positive association (positive correlation coefficient) from MPB to NT in every metropolitan area, indicating that an increase in MPB would translate into an increase of NT throughout the country (Table 2).

In three supplementary analyses, we ensured that our results are robust. First, we ensured the robustness of transfer entropy with respect to memory effects, by performing further computations using longer time histories that accounted for more than one time step in the past (see Supplementary Note 3). Second, we conducted transfer entropy analysis with time series of NT and PT generated with sentiment analysis using the classical natural language toolkit (NLTK) package of Bird et al.[96], which relies on the Vader dictionary[97] (see Supplementary Note 4 for details). Finally, motivated by a recent survey by Gallup[98] suggesting that people are capable of gauging local crime rates (in spite of the biased depiction of crimes by the media), we conducted analogous analyses by utilizing local crimes rather than their media coverage (see Supplementary Note 5 for details). In all three studies, the conclusions of our transfer entropy analysis are equivalent. These analyses were not part of the initial hypotheses of the study, yet, they help understand the perception of the police by the general public as they experience crime in their neighborhoods.

**Analysis through convergent cross mapping.** CCM was used to test the hypotheses put forward (H1a, H1b, and H2). Similar to transfer entropy, CCM captured the coupling between MPB and NT through all metropolitan areas studied (Fig. 6). With respect to H1a and H1b hypotheses, MLC was poorly reconstructed from the Twitter time series in most cases (except for Houston), thereby confirming the conclusions of the transfer entropy analysis. Although we registered non-converging predictions in New York, MPB was better reconstructed. Hence, it can be stated that even if H1a and H1b should not be rejected in a few cases, MPB is the stronger driver of public sentiment towards the police.

**Detection of latent variables.** Based on the results in Supplementary Note 3, the maximum lag was set to one and we focused

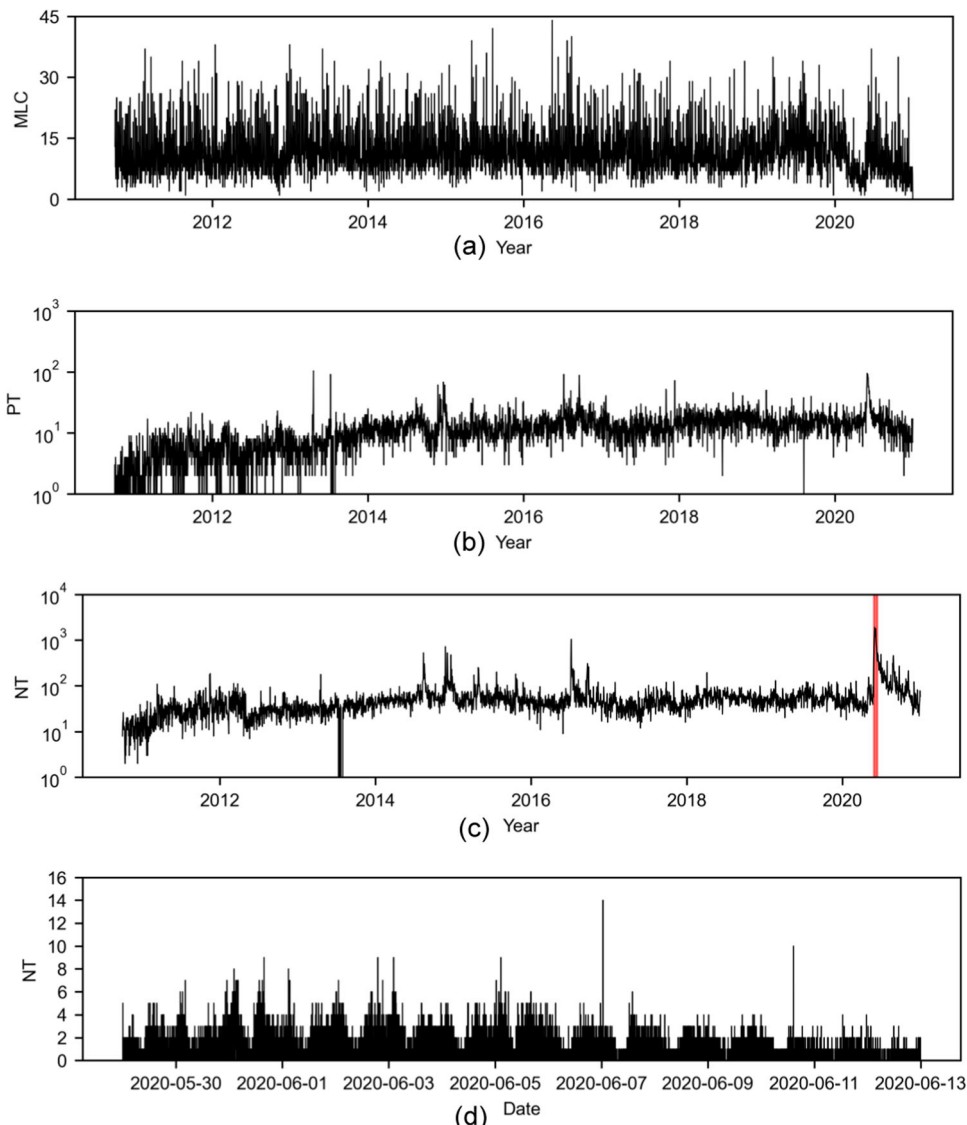

**Fig. 3 Time series for the New York City metropolitan area. a** Daily media coverage of local crime (MLC), with a peak of 44 articles registered on May 14, 2016. **b** Daily number of positive tweets (PT) about the police, with a peak of 104 registered on April 5, 2013. **c** Daily number of negative tweets (NT) about the police, with a peak of 1896 registered on May 31, 2020; activity in the wake of George Floyd's murder is highlighted in red. **d** Zoomed-in view at the resolution of one minute of the number of negative tweets about the police in the wake of George Floyd's murder period, from May 29, 2020 until June 13, 2020.

on the synchronous (within one day) association between MPB and NT. For nine of the 18 metropolitan areas, the algorithm excludes the presence of any latent variable that would confound the association from MPB to NT (Table 3). For seven metropolitan areas, the algorithm yields inconclusive predictions (a latent variable may or may not exist), for one metropolitan area (Los Angeles) the presence of a link was deemed unlikely, and only in the Minneapolis metropolitan area it finds evidence of a latent variable (Table 3). Which is the exact latent variable that simultaneously drove MPB and NT is not an output of LPCMCI. It could be argued that it is no chance that Minneapolis, the location of George Floyd's killing, is the singular metropolitan area in which a latent variable is present, but we have no means to pinpoint which latent variable may be in effect.

**High-resolution analysis in the wake of George Floyd's murder.** This analysis on the highly resolved time series (Fig. 3d and Supplementary Figs. 1–17) was performed with various widths of

the time window about the media post (Fig. 7a), ranging from 60 minutes (30 before and 30 after) to 1440 minutes (720 before and 720 after). Statistical comparison for all the metropolitan areas (one-tailed Wilcoxon signed-rank test, $n = 205$) indicates that for the vast majority of metropolitan areas (at least 12 out of 18 over time windows between 240 and 1080 minutes, Fig. 7b, c), there is an increase in the public's expression of negative attitude towards the police on Twitter in response to media coverage of police brutality by newspapers on Twitter. This independent analysis offers further support to H2, while indicating that the time scale for the public to respond to the media is more than one hour and it can be as much as almost half a day.

## Discussion
Mass media and social networks have been home to a growing debate regarding police brutality events[99–101], suggesting that media coverage of such events can shape public sentiment towards the police. Another factor that could potentially drive the

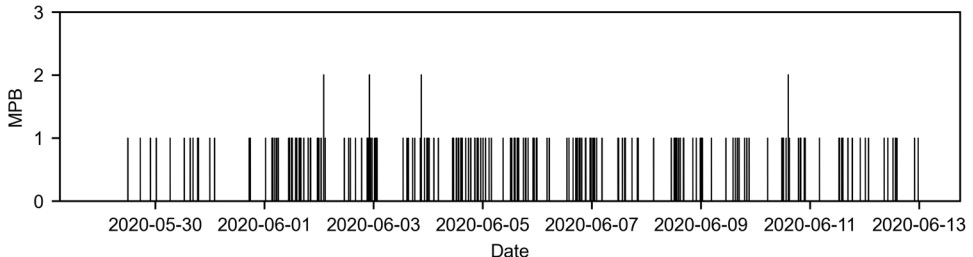

**Fig. 4 Aggregated statistics of public sentiment about the police.** Daily means of positive tweets (green, PT), negative tweets (red, NT), and media coverage of local crime (blue, MLC) in the chosen metropolitan areas. The error bars represent one standard deviation from the mean.

**Fig. 5 High-resolution time series of media coverage of police brutality after George Floyd's murder.** Number of tweets posted by the 10 most-followed Twitter accounts of newspapers mentioning "police brutality" in the wake of George Floyd's murder, from May 29, 2020 until June 13, 2020. A total of 209 tweets were recorded; the average time lapse between two consecutive tweets is 104.3 minutes.

**Table 1 Conditional transfer entropy results for the 18 chosen metropolitan areas.**

| Metropolitan area | MLC → NT \| MPB | | MLC → PT \| MPB | | MPB → NT \| MLC | |
|---|---|---|---|---|---|---|
| Atlanta | 0.0068 | (0.0099) | 0.0072 | (0.0099) | **0.0306** | (0.0099) |
| | $p = 0.5267$ | | $p = 0.4121$ | | $p < 0.0001$ | |
| Baltimore | 0.0054 | (0.0099) | **0.0105** | (0.0099) | **0.0189** | (0.0099) |
| | $p = 0.8397$ | | $p = 0.0273$ | | $p < 0.0001$ | |
| Boston | 0.0061 | (0.0099) | 0.0043 | (0.0099) | **0.0318** | (0.0099) |
| | $p = 0.6923$ | | $p = 0.9619$ | | $p < 0.0001$ | |
| Chicago | 0.0098 | (0.0099) | 0.0071 | (0.0099) | **0.0337** | (0.100) |
| | $p = 0.0554$ | | $p = 0.4306$ | | $p < 0.0001$ | |
| Dallas | **0.0115** | (0.0099) | 0.0077 | (0.0099) | **0.0375** | (0.0100) |
| | $p = 0.0084$ | | $p = 0.3059$ | | $p < 0.0001$ | |
| Denver | 0.0085 | (0.0099) | 0.0056 | (0.0099) | **0.0341** | (0.0099) |
| | $p = 0.1706$ | | $p = 0.7977$ | | $p < 0.0001$ | |
| Detroit | 0.0078 | (0.0099) | 0.0063 | (0.0099) | **0.0280** | (0.0099) |
| | $p = 0.3010$ | | $p = 0.6308$ | | $p < 0.0001$ | |
| Houston | 0.0095 | (0.0100) | 0.0075 | (0.0099) | **0.0205** | (0.0100) |
| | $p = 0.0781$ | | $p = 0.3425$ | | $p < 0.0001$ | |
| Los Angeles | 0.0082 | (0.0100) | 0.0064 | (0.0100) | **0.0326** | (0.0100) |
| | $p = 0.2309$ | | $p = 0.6182$ | | $p < 0.0001$ | |
| Miami | 0.0062 | (0.0099) | 0.0072 | (0.0099) | **0.0344** | (0.0099) |
| | $p = 0.6642$ | | $p = 0.4333$ | | $p < 0.0001$ | |
| Minneapolis | 0.0063 | (0.0099) | 0.0072 | (0.0099) | **0.0305** | (0.0099) |
| | $p = 0.6439$ | | $p = 0.4153$ | | $p < 0.0001$ | |
| New York | 0.0091 | (0.0100) | 0.0054 | (0.1000) | **0.0291** | (0.0100) |
| | $p = 0.1178$ | | $p = 0.8320$ | | $p < 0.0001$ | |
| Philadelphia | 0.0056 | (0.1000) | 0.0066 | (0.0099) | **0.0282** | (0.0099) |
| | $p = 0.7990$ | | $p = 0.5762$ | | $p < 0.0001$ | |
| Phoenix | 0.0064 | (0.0100) | 0.0073 | (0.0100) | **0.0388** | (0.0100) |
| | $p = 0.6149$ | | $p = 0.3887$ | | $p < 0.0001$ | |
| Riverside | 0.0077 | (0.0099) | 0.0067 | (0.0099) | **0.0217** | (0.0100) |
| | $p = 0.3293$ | | $p = 0.5370$ | | $p < 0.0001$ | |
| San Francisco | 0.0066 | (0.0100) | 0.0053 | (0.0099) | **0.0293** | (0.0100) |
| | $p = 0.5665$ | | $p = 0.8451$ | | $p < 0.0001$ | |
| Tampa | 0.0096 | (0.0099) | 0.0073 | (0.0099) | **0.0391** | (0.0100) |
| | $p = 0.0634$ | | $p = 0.3957$ | | $p < 0.0001$ | |
| Washington, D.C. | **0.0111** | (0.0100) | 0.0097 | (0.0099) | **0.0382** | (0.0100) |
| | $p = 0.0148$ | | $p = 0.0624$ | | $p < 0.0001$ | |

The top left number in each cell of the table represents the conditional transfer entropy value ($m = 3744$). Numbers in parentheses reflect the 95% quantile of a surrogate distribution obtained from a permutation test. The $p$-values indicate the significance of the same test. Bold values indicate a significant conditional transfer entropy at $\alpha = 0.05$.

formation of public sentiment towards the police is media coverage of local crime, whereby the public will express more or less appreciation for the police as a function of the extent they feel safe. While several studies investigated the influence of local crime on one's perception of the police, literature on the influence of media coverage of local crime is limited[23]—our study is the first to measure and investigate it directly.

In contrast to previous studies that employed surveys to assess the public's opinion towards the police at a single point in time[11,62], we employ a model-free, information-theoretic approach to study associations between media coverage of police brutality, media coverage of local crime, and public sentiment towards the police. We provide quantitative evidence for testing two hypotheses: H1) public sentiment towards the police (positive or negative) is influenced by media coverage of local crime, and H2) negative public sentiment towards the police is influenced by national media coverage of police brutality, which offers a proxy of police misconduct.

Our results do not offer evidence in favor of H1, whereby a role of media coverage of local crime on either positive (H1a) and negative (H1b) sentiment was not captured in almost any of the conditional transfer entropy analyses. The same conclusion was obtained by pursuing an alternative analysis with CCM, which tests a different notion of causality based on dynamical systems theory. This outcome is in agreement with the findings of Jackson

and Sunshine[102] and Jackson and Bradford[103]. We cannot exclude the possibility that our analyses were not able to capture a relationship and we acknowledge that failing to reject a null hypothesis of independence between two variables is not sufficient to argue for their independence. However, since we were able to successfully detect associations with media coverage of police brutality with both information-theoretic and dynamical systems tools that aim to unveil causality, it is tenable that media coverage of local crime is not as strong of a driver of public sentiment of the police as media coverage of police brutality.

While not part of our original set of hypotheses about media coverage, we conducted additional analyses using objective measures of local crimes (number of local crimes and severity-weighted number of local crimes). Results of additional analyses for New York City, where such a data is made available to the public, yielded similar conclusions to those obtained with their local media coverage. Although limited to the case of New York City, the similarity between causal inferences obtained with number of crimes, number of crimes weighted by their severity, and their media coverage suggests that the public sentiment towards the police is only marginally affected by local crimes (whether gauged directly by residents or apprehended through the media). The absence of a causal effect of local crimes on public sentiment may find a basis in negativity bias theory, which states that humans tend to make sense of their environment based

**Table 2 Partial correlation results for the 18 chosen metropolitan areas.**

| Metropolitan area | $\rho$ | 95% confidence interval | p-value |
|---|---|---|---|
| Atlanta | 0.1982 | [0.17, 0.23] | $p < 0.0001$ |
| Baltimore | 0.1714 | [0.14, 0.20] | $p < 0.0001$ |
| Boston | 0.2139 | [0.18, 0.24] | $p < 0.0001$ |
| Chicago | 0.2155 | [0.18, 0.25] | $p < 0.0001$ |
| Dallas | 0.1901 | [0.16, 0.22] | $p < 0.0001$ |
| Denver | 0.2220 | [0.19, 0.25] | $p < 0.0001$ |
| Detroit | 0.2015 | [0.17, 0.23] | $p < 0.0001$ |
| Houston | 0.1349 | [0.10, 0.17] | $p < 0.0001$ |
| Los Angeles | 0.2044 | [0.17, 0.23] | $p < 0.0001$ |
| Miami | 0.2209 | [0.19, 0.25] | $p < 0.0001$ |
| Minneapolis | 0.2099 | [0.18, 0.24] | $p < 0.0001$ |
| New York City | 0.2154 | [0.18, 0.25] | $p < 0.0001$ |
| Philadelphia | 0.2202 | [0.19, 0.25] | $p < 0.0001$ |
| Phoenix | 0.2117 | [0.18, 0.24] | $p < 0.0001$ |
| Riverside | 0.1892 | [0.16, 0.22] | $p < 0.0001$ |
| San Francisco | 0.2200 | [0.19, 0.25] | $p < 0.0001$ |
| Tampa | 0.2843 | [0.25, 0.31] | $p < 0.0001$ |
| Washington, D.C. | 0.2012 | [0.17, 0.23] | $p < 0.0001$ |

The partial correlation and 95% confidence interval between media coverage of police brutality at time $t-1$ and number of negative tweets at time $t$ was performed to infer the sign of their association (m = 3,744). The $\rho$ is the Spearman partial correlation coefficient and the $p$-values correspond to the null hypothesis of $\rho = 0$.

on negative information rather than positive information[104]. For example, Li et al.[105] and Miller et al.[106] showed that a negative encounter between citizens and the police has a stronger influence on their attitude towards the police than a positive encounter. In this vein, people would react less to successful crime prevention by law enforcement officers than they would to reports of police brutality.

In support of H2, our results revealed that media coverage of police brutality could be a key driver of negative sentiment of the public towards the police, whereby increased media coverage of police brutality affects the volume of the public discourse on Twitter (increased number of negative tweets). Furthermore, in agreement with the literature suggesting that news of police brutality reverberates throughout the country[11], we unraveled the role of media reporting of police brutality at a national level. That is, all urban populations responded to instances of police misconduct on social media, even if the incidents took place elsewhere. Were the effect of police brutality on public sentiment towards the police only local (residents responding only to police brutality in their city), our analysis would unlikely yield a link from national media coverage to public sentiment. In such a hypothetical case, aggregating media at a national level would have "diluted" the causal role of the media and mitigated associations in some of the 20 metropolitan areas, in contrast with our findings.

Our finding is in line with the observations made by Miller et al.[106], who proposed that media coverage of police routine has a limited effect on the public's opinion towards the police, in contrast with reporting of misconduct and vicarious experiences of police abuse that shape our view of law enforcement. Such a response is also in agreement with previous accounts that found negativity to spread more than positivity on Twitter. For example, Schöne et al.[107] demonstrated that negative language regarding political matters tends to spread more than positive language; likewise, Hansen et al.[108] determined that the extent to which news become viral is affected by their negative content, such negative news content is more likely to be re-tweeted. Although our focus is on the appraisal of information through social media

rather than real interactions between people and the police, there are similarities between our findings and the literature on the role of personal interactions between citizens and police. In particular, personal interactions with the police can leave a profound mark on citizens' opinion about the police. For example, Skogan[109] surveyed 3005 citizens in Chicago and showed that a bad experience with the police significantly affects the public's assessment of the police. Likewise, in controlled experiments, Mazerolle et al.[110] and Weisburd et al.[111] showed that policing that is carried out with dignity and respect is conducive to cooperation with citizens and may even lead to a reduction in crime incidents. Not only will negative interactions with the police hinder one's confidence in the police, but they also spread to family and friends[106]. The role of cultural intelligence and other intercultural models on police-community work are reviewed by Louis and Grantham[112].

Transfer entropy is based on the premise that causal associations do not instantaneously unfold, such that only the past of a source variable will contribute to the present uncertainty of a target variable. Such a condition is difficult to guarantee when working with Twitter data, whereby it is tenable that a sizable portion of the public will respond within the same day to a breaking news on police brutality[113]. To better detail the process by which the public reacts to these news, we performed an independent analysis at the resolution of one minute in the two-week period following the murder of George Floyd (starting four days after his death). In agreement with transfer entropy conclusions, we found a significant increase in the number of negative tweets from the public about the police following a tweet from a newspaper regarding police brutality across most of the metropolitan areas. Over the chosen two weeks, during which George Floyd's death had become known to most of the country[5], the public reacted to the media by expressing their negative sentiment to news that were being posted on Twitter accounts of newspapers. Had the association between media coverage of police brutality and negative public sentiment about the police been an artifact of the transfer entropy analysis, we should have not registered a clear sequence of posting events like the one we demonstrated; rather, posts of newspapers and general users should have been independently spaced in time. Even during this dire period of discourse on police brutality, the time scale of public response was on the order of several hours, thereby offering indirect backing to the use of daily data in the transfer entropy analysis for our observation window of more than 10 years.

Interestingly, the same behavior was observed throughout all the metropolitan areas under investigation. That is, irrespective of political leaning, population density, police budget, and other socio-demographic characteristics, people tweeted negative sentiment about the police following the outbreak of news about police brutality, paying limited attention to media about local crime. The prevalence of this consistent pattern throughout the US points at some key similarities in how we, as a society, appraise law enforcement. We cannot argue that such a consistency was expected at the beginning of our study, given the wide variation of human response to controversial, political topics similar to police brutality, such as the responses to the recent COVID-19 pandemic[114] and gun control[115].

We presented multiple direct and indirect evidences that latent variables are unlikely to play a role on the observed associations. First, the analysis with LPCMCI[65] showed that, for the vast majority of metropolitan areas, there are no latent variables that would confound the causal link from police brutality to negative tweets. Second, in CCM, the values of $\rho$ (Fig. 6) were as high as 90% for the majority of the metropolitan areas, indicating that

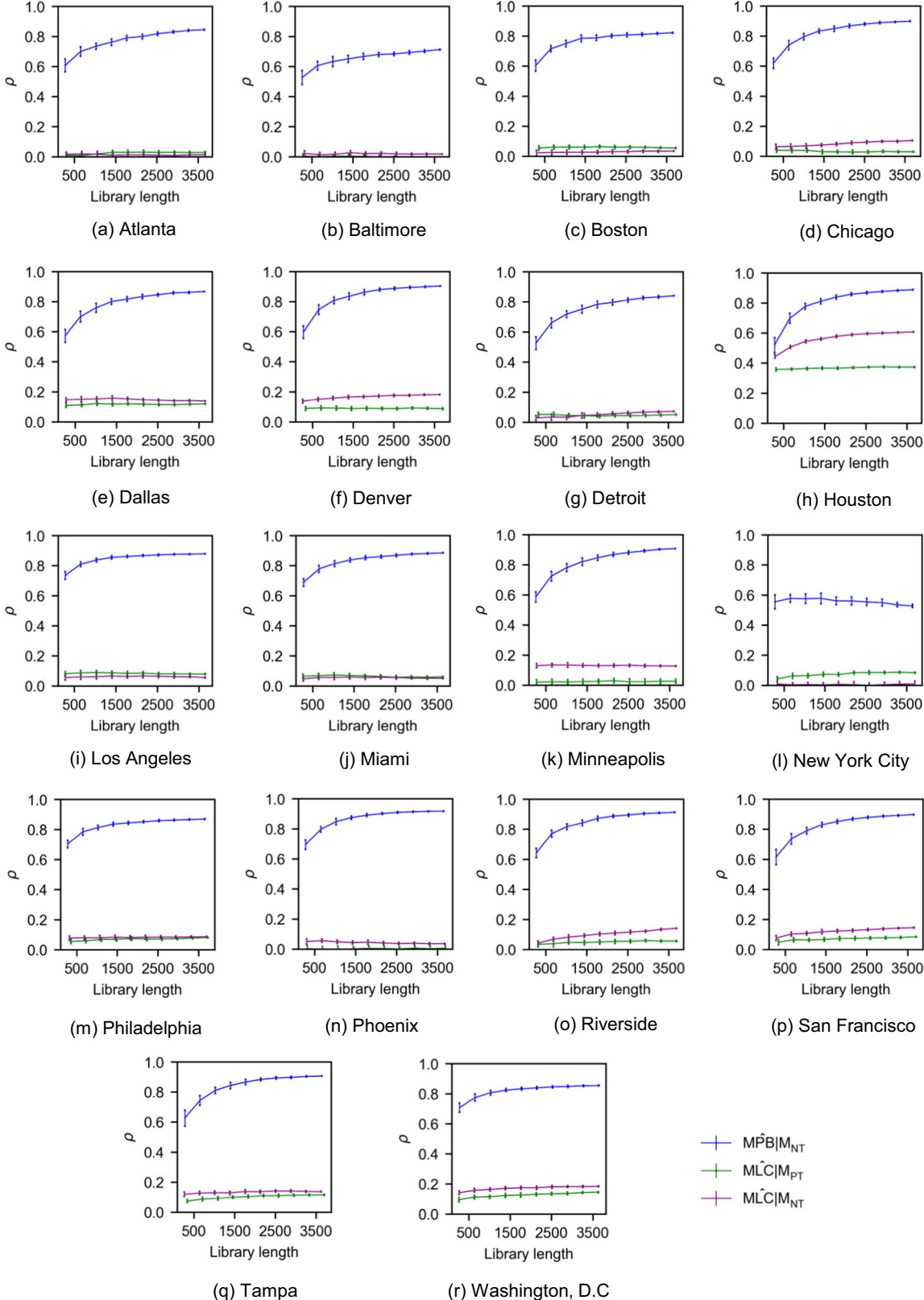

**Fig. 6 Convergence plots of CCM for the 18 metropolitan areas** . Each line corresponds to one of the four hypotheses (H1a, H1b, and H2). For each library length, a total of 30 estimations were performed. The error bars represent one standard deviation of the Pearson correlation coefficient estimated for each library length.

**Table 3 Analysis of the causal relationship between media coverage of police brutality (MPB) and number of negative tweets (NT) through the LPCMCI algorithm for all 18 metropolitan areas ($m = 3745$).**

| Metropolitan area | Link type | $\rho$ | 95% confidence interval | p |
|---|---|---|---|---|
| **Atlanta** | MPB → NT | 0.1299 | [0.0983, 0.1613] | < 0.0001 |
| Baltimore | MPB * − * NT | 0.1294 | [0.0978, 0.1608] | < 0.0001 |
| **Boston** | MPB → NT | 0.1782 | [0.1470, 0.2091] | < 0.0001 |
| Chicago | MPB ∘ → NT | 0.2005 | [0.1696, 0.2311] | < 0.0001 |
| **Dallas** | MPB → NT | 0.1600 | [0.1286, 0.1910] | < 0.0001 |
| **Denver** | MPB → NT | 0.1793 | [0.1481, 0.2101] | < 0.0001 |
| Detroit | MPB ∘ − ∘ NT | 0.1641 | [0.1328, 0.1951] | < 0.0001 |
| **Houston** | MPB → NT | 0.0958 | [0.0640, 0.1275] | 0.0002 |
| Los Angeles | MPB ← ∘ NT | 0.1016 | [0.0698, 0.1332] | < 0.0001 |
| **Miami** | MPB → NT | 0.2019 | [0.1710, 0.2325] | < 0.0001 |
| Minneapolis | MPB $\overset{1}{\leftrightarrow}$ NT | −0.0404 | [−0.0723, −0.0083] | 0.0135 |
| **New York** | MPB → NT | 0.1269 | [0.0953, 0.1583] | < 0.0001 |
| **Philadelphia** | MPB → NT | 0.2145 | [0.1837, 0.2448] | < 0.0001 |
| Phoenix | MPB ∘ − ∘ NT | 0.1123 | [0.0805, 0.1438] | < 0.0001 |
| Riverside | MPB * − * NT | 0.1545 | [0.1230, 0.1856] | < 0.0001 |
| San Francisco | MPB * − * NT | 0.1733 | [0.1420, 0.2042] | < 0.0001 |
| **Tampa** | MPB → NT | 0.1315 | [0.0999, 0.1629] | < 0.0001 |
| Washington, D.C | MPB * − * NT | 0.1760 | [0.1447, 0.2068] | < 0.0001 |

The first column specifies the metropolitan area. The second column defines the link type: $X \rightarrow Y$ implies that $X$ is a cause of $Y$ and $Y$ does not cause $X$; $X^* − ^*Y$ implies a conflict in the algorithm that leads to an inconclusive conclusion; $X \leftrightarrow Y$ implies that $X$ does not cause $Y$, $Y$ does not cause $X$, and $X$ and $Y$ are subject to unobserved confounding (there is a latent variable $Z$ that causes both $X$ and $Y$); $X \circ − \circ Y$ implies that $X$ may or may not cause $Y$ and $Y$ may or may not cause $X$; and $X \circ \rightarrow Y$ implies that $X$ may or may not cause $Y$ and $Y$ does not cause $X$. The number above the link corresponds to the lag of the link, where the absence of a number implies a contemporaneous one. In case of multiple links, we report the link with the lowest lag. The third column is Pearson's $\rho$ and the fourth is the $p$-value of conditional independence between the variables (all values are below the significance level $\alpha = 0.05$). For clarity, the names of the nine metropolitan areas in which LPCMCI uncovers a causal link from MPB to NT are highlighted in bold font. Confidence intervals were computed manually, using Scipy's stats Pearson's r confidence interval, as Tigramite does not output the confidence intervals for LPCMCI.

media coverage of police brutality drives Twitter response nearly perfectly and ruling out the presence of any latent variable[116]. Finally, the high-resolution minute-by-minute analysis around the time of George Floyd's murder offers compelling evidence that, throughout the country, people responded to news about police brutality on Twitter within a few hours, leaving limited room for alternative explanations in terms of latent variables. Overall, these results support the notion that a causal effect of media coverage of police brutality on the number of negative tweets discovered through multiple means (transfer entropy, CCM, and high-resolution, minute-by-minute analysis) is not an artifact of latent variables, but rather an instance of direct causation.

**Limitations**. The data acquisition and analysis of this study are not free of limitations. In particular, we identify five main limitations.

1. Sentiment analysis of Twitter data may not accurately represent the sentiment of the entire public. Certain sectors of the population may not use this social media platform and among those who use Twitter, likely not all express their views towards the police through tweets[117]. It is also tenable that a number of potentially useful tweets were not collected in this study as they were not geo-located. Although performed with state-of-the-art technique and validated with two different methods, we acknowledge that sentiment analysis is susceptible to noise in the detection of slang words, icons, and sarcastic expressions[118].

2. The conclusions of the study may not generalize to every urban community, as we limited our analysis to large metropolitan areas that may experience more policing[119]. Differences in the exposure to police work may lead to different attitudes towards the police[120]. On top of that, people's appraisal of the police may be shaped by their specific experience with law enforcement in the past[106,109–111], as well as socio-demographic factors[121].

For example, it is known that legal cynicism and distrust in the police are higher in disadvantaged neighborhoods that are susceptible to crimes[13,14]. These measures cannot be disentangled from media coverage of police brutality or crime and their consumption by those communities. For example, Laniyonu[122] showed that people's association with a cultural, ethnic, or racial group (black diaspora, in this instance) may instigate a stronger sentiment within those communities towards the police in response to brutality incidents.

3. Our analysis does not allow for teasing out the specific role of media coverage of local police brutality events on public sentiment. Due to the infrequency of these events, causal inference is difficult to perform. A transfer entropy analysis on time series with such a low information content may be prone to false inferences[75].

4. The notion of causality that was discussed throughout the paper should be interpreted with caution as we only investigated observational data with a method that is not free of assumptions. Even though we overcame many of the limitations of transfer entropy by applying different causal inference techniques, these methods still have their own assumptions and limitations[123]. For example, LPCMCI assumes faithfulness, no selection bias, and acyclic causality in the time-series graphs, and it does not rule out the possibility of indirect causal links. For example, the association between media coverage of police brutality and public sentiment could be mitigated by the police's own response to media coverage of police brutality; Through this lens, the police would react to news about police brutality and change their behavior accordingly, whether by adoption of new policies or by natural reaction of the officers to the news. In turn, this response could affect the public's sentiment. While the high-resolution analysis in the wake of George Floyd's murder hints at a direct effect due to the fast response, the aforementioned mechanism may not be ruled out as a general causal mechanism for longer reaction

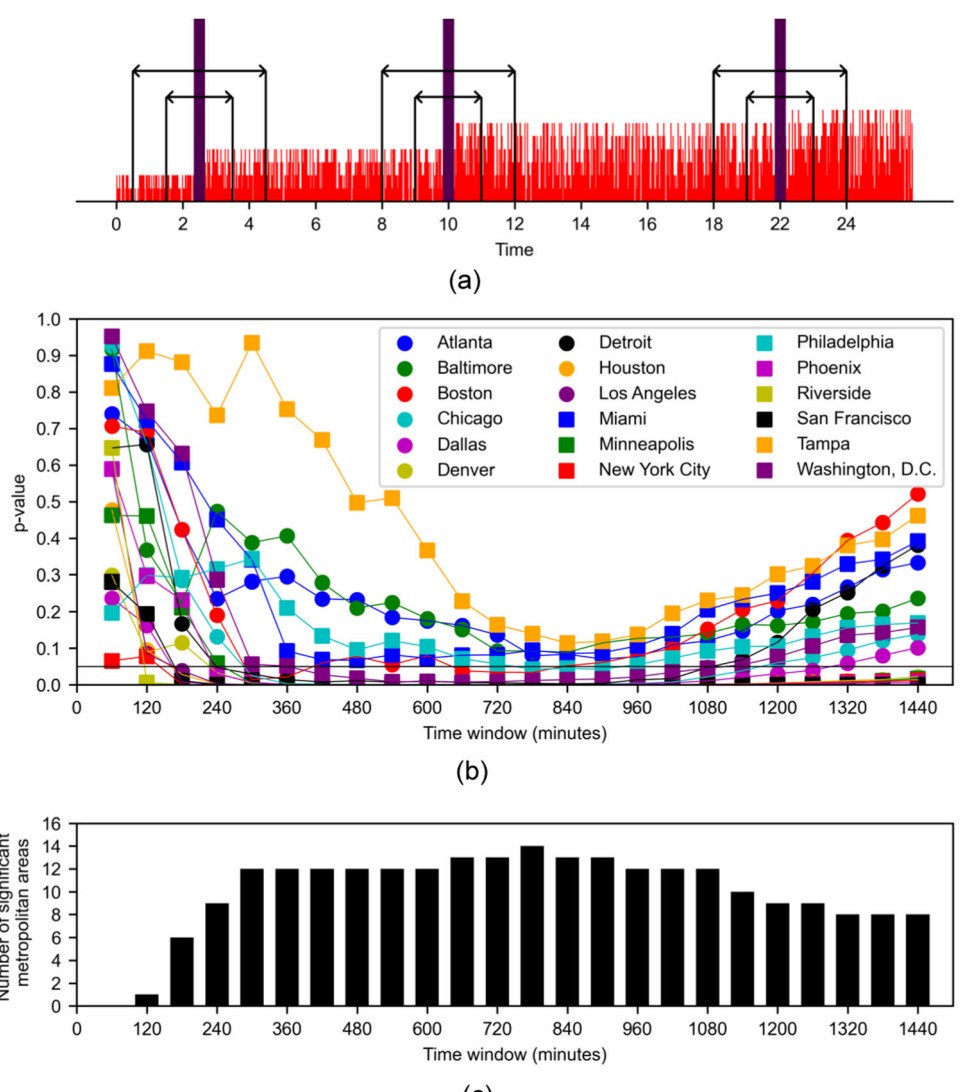

**Fig. 7 High-resolution analysis in the wake of George Floyd's murder. a** Sketch of the procedure used in the analysis; an artificial time series is utilized for better visualization. Purple thick vertical lines represent a tweet about police brutality from a newspaper and red lines are the numbers of negative tweets by the public. Smaller arrows identify 120-min time window (60 before and 60 after) and larger arrows 240-min period (120 before and 120 after), during which average numbers of negative tweets are calculated. **b** *P*-values of one-tailed Wilcoxon signed-rank test (*n* = 205) as a function of the size of the chosen time window around the posting of a tweet about police brutality by newspapers. The test compares the average number of negative tweets per minute about the police that are posted by the public before and after the posting of a media tweet mentioning "police brutality". The horizontal black line represents the chance at a significance level $\alpha = 0.05$. The null hypothesis is that there is no difference between the means of the distributions of the average numbers of negative tweets by the public before and after a post in a newspaper on police brutality. **c** Number of metropolitan areas for which the Wilcoxon test rejects the null hypothesis at $\alpha = 0.05$ as a function of the size of the chosen time window around the posting of a tweet about police brutality by newspapers -- number of metropolitan areas below the significance threshold in (**b**).

time, calling for similar research involving police behavior data.

5. While our results feature strong statistical significance, we warn care with respect to sampling of sentiment across the political aisle, whereby the composition of Twitter users in each metropolitan area may not fully replicate the make-up of Republican and Democratic voters at any point in time. The data presented in Yeung et al.[124] point to a representation of Democratic and Republican affiliations on Twitter in 2020 (56% and 44%, respectively) that is consistent with the party affiliation of US citizens reported in Gallup[125] data of the same period (52% and 48%). In contrast, the survey conducted by Pew Research Center between November and December 2018 (around mid-term elections) suggests an over-representation of Democratic

users on Twitter[117]. Such differences may introduce a bias to the measure of public sentiment in a study, which varies by political ideology and affiliation[5]. Beyond the political composition of Twitter users, another factor to consider is the extent to which tweets by Democrats and Republicans vary in their positivity or negativity.

## Conclusions
This study provides quantitative evidence of directional interdependencies between media coverage of local crime, media coverage of police brutality, and public sentiment towards the police, expanding on the state of knowledge that has largely relied on correlation analyses and descriptive statistics on single response, "snapshot" surveys[126]. This study examines the

complete ecosystem of public-police interaction rather than some of its individual components. Through its original, information-theoretic approach, this study confirms our prediction of a strong influence of media coverage of police brutality on the public's sentiment towards the police. At the same time, media reports of local crime, as well as local crimes, were not identified as salient drivers of public sentiment. This observation is likely to be rooted deep in our society, as recognized by Block[127] 50 years back, when he wrote "citizen support for the police is constructed out of good and respectful policework. The negative effect of fear of the police on support for the police was far stronger than the positive effect of fear of crime". This notion brings to question a need for mechanisms to highlight the contributions of the police to society in addition to publicizing their misconduct, towards a debate of police reformation that is less biased and based on a holistic viewpoint of the role and need of law enforcement. Given the critical weight of negative news, media coverage should consider deliberate efforts towards reporting a more balanced projection of police-related news.

The study of public opinion is a long venture that can take on many forms. Childs[128] writes about 50 different definitions of public opinion. Taking a further step, Noelle[129] discusses the lack of consensus on a common operational definition and suggests abandoning the phrase "public opinion" entirely. This study does not seek to shed light on how people might rationally assess the political topics around the police and form an opinion about them. Rather, it tackles the spontaneous, impulsive, and emotional response of the public towards the actions of law enforcement, one of the many aspects of public opinion that bears a critical role in our hyper-connected society consuming and creating knowledge through mass and social media.

Violence and abuse of power seriously damage police reputation in the eyes of the public. In the context of urbanization, addressing police brutality and mitigating police-community tensions become even more critical. As cities become larger, they experience more crimes per capita at a super-linear scale[130]. In stark contrast, the size of the police force and budget that scale sublinearly in bigger cities[130]. That is, larger cities have less police funding and patrolling per resident. Given that events of police brutality stem from police officers' fear of crime and injury[131,132], we anticipate a bidirectional amplification of a negative trend where police and community grow further and further apart.

## Data availability

Partial data needed to evaluate the conclusions in the paper are available on Github[133]. Raw Twitter data can not be shared, hence only the processed time series are made publicly available.

## Code availability

All codes needed to evaluate the conclusions in the paper are available on Github[133].

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

## Acknowledgements

This study was supported by the National Science Foundation under award CMMI-1953135. S.R. acknowledges the support of a Spain-US Fulbright grant co-sponsored by Fundación Séneca and of Ministerio de Ciencia e Innovació of Spain under grant PID2019-107192GB-I00. R.B.V. was supported in part through a postdoctoral award from The National Collaborative on Gun Violence Research. The funders had no role in study design, data collection and analysis, and decision to prepare or publish the manuscript. The views expressed in this article are the authors' and do not necessarily reflect the view of the funders. The authors would like to express their gratitude to the NYU CUSP Capstone team "Understanding public opinion towards the police in New York City" (Ruoqing Lin, Lingxuan Bu, and Xiangyu Ying), who helped in data collection. This work was supported in part through the NYU IT High Performance Computing resources, services, and staff expertise.

## Author contributions

Conceptualization—R.S., S.R., M.P.; methodology—R.S., S.R., M.P.; software—R.S., S.R., M.P.; formal analysis—R.S., S.R., M.P.; resources—M.P.; data curation—R.S., S.R.; writing—original draft preparation—R.S., S.R., R.D., R.B.V., M.P.; writing-review and editing—R.S., S.R., R.D., R.B.V., M.P.; visualization—R.S., S.R.; supervision—M.P.; funding acquisition—S.R., R.B.V., M.P.

## Competing interests

The authors declare no competing interests.
