## [Peer Review File · Communications Psychology]

1st Dec 23

Dear Mau,

Thank you for your patience during the peer-review process. Your manuscript titled "Understanding the role of media in the formation of public sentiment towards the police" has now been seen by 2 reviewers, and I include their comments at the end of this message. One of the reviewers, Reviewer #4 is the same Reviewer #4 who vetted your work previously. Reviewer #6 is a new referee. Both reviewers have subject-matter expertise, and Reviewer #4 is also an expert in the statistical analysis.

The reviewers find your work of interest and improved over previous versions, but raised some important points. We are interested in the possibility of publishing your study in Communications Psychology, but would like to consider your responses to these concerns and assess a revised manuscript before we make a final decision on publication.

We therefore invite you to revise and resubmit your manuscript, along with a point-by-point response to the reviewers. Please highlight all changes in the manuscript text file.

Editorially, we require you to significantly tone down all claims of causality. Your analysis relies on observational data, and as the reviewers highlight, there are multiple concerns that may affect the data and thus, regardless of the analysis, weaken causal inference. Please remove the causal claims from the Abstract, Introduction, and from the subheadings. In the Discussion, you may explain why you consider the findings to indicate causality, but these statements must be caveated. The discussion of limitations should be comprehensive and appear under its own subheading.

Please also include detailed information on when the Twitter data were downloaded (date range), whether the data were obtained through a dedicated AIP, and include a note on the ToS underlying the approach.

To facilitate the revisions and ensure that future processing toward potential acceptance runs faster, I provide a detailed checklist in the attachment. Please ensure you respond to every item by implementing suitable changes in the manuscript. You may also use the right-hand column to confirm that changes have been implemented.

Please use the following link to submit your revised manuscript, point-by-point response to the referees' comments (which should be in a separate document to any cover letter), and the completed checklist:

[link redacted]

We hope to receive your revised paper within 8 weeks; please let us know if you aren't able to

submit it within this time so that we can discuss how best to proceed. If we don't hear from you, and the revision process takes significantly longer, we may close your file. In this event, we will still be happy to reconsider your paper at a later date, provided it still presents a significant contribution to the literature at that stage.

Please do not hesitate to contact me if you have any questions or would like to discuss these revisions further. We look forward to seeing the revised manuscript and thank you for the opportunity to review your work.

Best wishes,

Marike

Marike Schiffer, PhD
Chief Editor
Communications Psychology

EDITORIAL POLICIES AND FORMATTING

We ask that you ensure your manuscript complies with our editorial policies.

Editorial Policy: <https://www.nature.com/documents/nr-editorial-policy-checklist.pdf> Policy requirements (Download the link to your computer as a PDF.)

* **CODE AVAILABILITY:** All Communications Psychology manuscripts must include a section titled "Code Availability" at the end of the methods section. In the event of publication, we require that the custom analysis code supporting your conclusions is made available in a publicly accessible repository; at publication, we ask you to choose a repository that provides a DOI for the code; the link to the repository and the DOI will need to be included in the Code Availability statement. Publication as Supplementary Information will not suffice. We ask you to prepare code at this stage, to avoid delays later on in the process.

* **DATA AVAILABILITY:**

All Communications Psychology manuscripts must include a section titled "Data Availability" at the end of the Methods section or main text (if no Methods). More information on this policy, is available at <http://www.nature.com/authors/policies/data/data-availability-statements-data-citations.pdf>

data-citations.pdf.

At a minimum the Data availability statement must explain how the data can be obtained and whether there are any restrictions on data sharing. Communications Psychology strongly endorses open sharing of data. If you do make your data openly available, please include in the statement:

We recommend submitting the data to discipline-specific, community-recognized repositories, where possible and a list of recommended repositories is provided at http://www.nature.com/sdata/policies/repositories.

If a community resource is unavailable, data can be submitted to generalist repositories such as figshare or Dryad Digital Repository. Please provide a unique identifier for the data (for example a DOI or a permanent URL) in the data availability statement, if possible. If the repository does not provide identifiers, we encourage authors to supply the search terms that will return the data. For data that have been obtained from publicly available sources, please provide a URL and the specific data product name in the data availability statement. Data with a DOI should be further cited in the methods reference section.

Please refer to our data policies at http://www.nature.com/authors/policies/availability.html.

REVIEWERS' COMMENTS:

Reviewer #4 (Remarks to the Author):

In this latest version of the paper (relative to earlier submission that I reviewed in a sister journal), the authors have made several improvements. They have altered their hypotheses to better reflect statements about public discourse on Twitter rather than how the public's perception itself will respond to media coverage of local crimes and police misconduct. They have adjusted their sentiment analysis approach to employ more sophisticated models that use aspect-based sentiment analysis. They have added an additional approach to deal with confounding variables using LPCMCI. I comment on these in more detail below.

The altered hypotheses are more in line with the authors' analysis. It is worth noting that the changes in the hypotheses are meaningful in terms of what the authors can conclude based on their analysis. It is worth mentioning that, even if we believe Twitter users are representative of the general public, it is not clear whether their tweets are representative of how the public responds to

media coverage of local crime and police misconduct. Some Twitter users may be more or less willing to speak out about issues on Twitter. In this way, tweets may not accurately reflect public opinion on sensitive topics, particularly when those topics pertain to race relations. The authors now acknowledge this limitation explicitly in their discussion.

The change to aspect-based sentiment is a welcome improvement, as overall sentiment of posts may be misleading relative to aspect-based sentiment with respect to the focal terms “police” and “cop”. It is easy to imagine how the former approach could misconstrue sentiment relative to the latter approach.

Overall, I am still wary of the authors’ attempt to estimate these causal relationships. It is certainly true that the approaches the authors employed overcome some of the limitations of classical correlational analysis. Given the observational nature of the study and the temporal resolution of the data, the recent changes that aim at mitigating potential confounds due to latent observables are perhaps the best that one could expect, and I think the authors have done a much better job of frankly acknowledging the limitations of these methods. However, there are still some other limitations that I think the authors need to do a better job of acknowledging, particularly in terms of LPCMCI. Their analysis is limited by lags -- they considered including an additional lag step for transfer entropy, but say they cannot go beyond this due to binning; however, it is plausible that confounds existing over longer lag periods that could in principle be discovered via the conditional independence tests of LPCMCI, if not for binning limitations. I think a bit more detail about the LPCMCI approach in the main paper (rather than in the supplemental materials alone) is warranted. Additionally, they should discuss the limitations of LPCMCI including the assumptions of no cyclic causal relationships, no selection bias, and the causal faithfulness assumption.

I would like to give one example of a confounder that I think their analysis does not rule out. Suppose we believe that the public reacts to local crime and police brutality that they are exposed to via mass media (TV, newspaper) and social media. But it seems likely that the police also react to this and alter their policing behaviors. This reaction is likely to be as contemporaneous as the public’s reaction to the former and may impact the public (through their direct interaction with the police) and thereby public discourse on Twitter as well. One could wrap up this mechanism up into the overall impact of media on the public, but doing so could ignore potentially different policy implications. For example, if the police response to such media reports is a major driver public sentiment toward the police, training that specifically targets how the police interact with the public after such well-publicized events might be called for. Could the authors comment on this?

Reviewer #6 (Remarks to the Author):

This article is a novel approach to examining the issue of whether community sentiment is influenced by media crime reporting or reporting on cases of officer brutality. The authors are correct when they state that this is a difficult endeavor as there are many variables that may influence the public's perception of the police. The authors took a novel statistical approach applying a method used in other professions, but relatively unheard of in the criminal justice field. The authors did a thorough job of explaining the reasoning behind using this statistical approach and walked the reader through the analysis step by step giving a clear overview and show the empirical support for each of their statistical decisions. The only question I would have and I don't think it

would influence the outcome, but did the authors take into consideration that there would be a power few tweeters? For example, in one day of 100 tweets, it was not 100 people tweeting, rather it was 80 people tweeting but 5 people tweet more than their peers. IDK if this would have an affect on the analysis, but there is research that supports the power few theory within most datasets. Outside this one issue, if the authors do not feel that this would affect their analysis, then it can be ignored. This paper is well written, is a novel statistical approach to an important topic that deserves attention and proper evaluation. I do not think the authors missed any important empirical work. The article is clear, concise, and thorough. This paper is ready for publication.

Reviewer #4

In this latest version of the paper (relative to earlier submission that I reviewed in a sister journal), the authors have made several improvements. They have altered their hypotheses to better reflect statements about public discourse on Twitter rather than how the public's perception itself will respond to media coverage of local crimes and police misconduct. They have adjusted their sentiment analysis approach to employ more sophisticated models that use aspect-based sentiment analysis. They have added an additional approach to deal with confounding variables using LPCMCI. I comment on these in more detail below.

The altered hypotheses are more in line with the authors' analysis. It is worth noting that the changes in the hypotheses are meaningful in terms of what the authors can conclude based on their analysis. It is worth mentioning that, even if we believe Twitter users are representative of the general public, it is not clear whether their tweets are representative of how the public responds to media coverage of local crime and police misconduct. Some Twitter users may be more or less willing to speak out about issues on Twitter. In this way, tweets may not accurately reflect public opinion on sensitive topics, particularly when those topics pertain to race relations. The authors now acknowledge this limitation explicitly in their discussion.

The change to aspect-based sentiment is a welcome improvement, as overall sentiment of posts may be misleading relative to aspect-based sentiment with respect to the focal terms "police" and "cop". It is easy to imagine how the former approach could misconstrue sentiment relative to the latter approach.

Response: Thank you for the constructive criticism you previously provided that helped us shape these new changes.

Overall, I am still wary of the authors' attempt to estimate these causal relationships. It is certainly true that the approaches the authors employed overcome some of the limitations of classical correlational analysis. Given the observational nature of the study and the temporal resolution of the data, the recent changes that aim at mitigating potential confounds due to latent observables are perhaps the best that one could expect, and I think the authors have done a much better job of frankly acknowledging the limitations of these methods. However, there are still some other limitations that I think the authors need to do a better job of acknowledging, particularly in terms of LPCMCI. Their analysis is limited by lags -- they considered including an additional lag step for transfer entropy, but say they cannot go beyond this due to binning; however, it is plausible that confounds existing over longer lag periods that could in principle be discovered via the conditional independence tests of LPCMCI, if not for binning limitations. I think a bit more detail about the LPCMCI approach in the main paper (rather than in the supplemental materials alone) is warranted. Additionally, they should discuss the limitations of LPCMCI including the assumptions of no cyclic causal relationships, no selection bias, and the causal faithfulness assumption.

Response: Thank you for the suggestion. We moved the entire LPCMCI analysis to the main manuscript and further expanded its assumptions in the new Limitations section. We also toned down causal claims throughout.

I would like to give one example of a confounder that I think their analysis does not rule out. Suppose we believe that the public reacts to local crime and police brutality that they are exposed

to via mass media (TV, newspaper) and social media. But it seems likely that the police also react to this and alter their policing behaviors. This reaction is likely to be as contemporaneous as the public's reaction to the former and may impact the public (through their direct interaction with the police) and thereby public discourse on Twitter as well. One could wrap up this mechanism up into the overall impact of media on the public, but doing so could ignore potentially different policy implications. For example, if the police response to such media reports is a major driver public sentiment toward the police, training that specifically targets how the police interact with the public after such well-publicized events might be called for. Could the authors comment on this?

Response: Thank for raising this important point. The mechanism you propose is certainly plausible, and we now discuss it in the Limitations section.

Reviewer #6

This article is a novel approach to examining the issue of whether community sentiment is influenced by media crime reporting or reporting on cases of officer brutality. The authors are correct when they state that this is a difficult endeavor as there are many variables that may influence the public's perception of the police. The authors took a novel statistical approach applying a method used in other professions, but relatively unheard of in the criminal justice field. The authors did a thorough job of explaining the reasoning behind using this statistical approach and walked the reader through the analysis step by step giving a clear overview and show the empirical support for each of their statistical decisions. The only question I would have and I don't think it would influence the outcome, but did the authors take into consideration that there would be a power few tweeters? For example, in one day of 100 tweets, it was not 100 people tweeting, rather it was 80 people tweeting but 5 people tweet more than their peers. IDK if this would have an affect on the analysis, but there is research that supports the power few theory within most datasets. Outside this one issue, if the authors do not feel that this would affect their analysis, then it can be ignored. This paper is well written, is a novel statistical approach to an important topic that deserves attention and proper evaluation. I do not think the authors missed any important empirical work. The article is clear, concise, and thorough. This paper is ready for publication.

Response: Thank you for your kind words and for the welcomed suggestion. We also believe that the Power Few effect does not change the results, but for the sake of completeness, we re-ran the transfer entropy statistical tests for H2 (media coverage of police brutality has an effect on public sentiment towards the police). In this new analysis, instead of the number of negative tweets per day, we used a weighted number of users who tweeted on a certain day as a measure of public sentiment. For example, if user A tweeted one negative tweet and user B tweeted one neutral tweet and one negative tweet. The weighted number of users who tweeted negatively about the police is 1.5 (1+1/2). The results of hypothesis 2 are summarized in the table below where we report the transfer entropy values and their corresponding p-values for each of the 18 cities. They are in agreement with our original analysis, demonstrating that the Power Few effect plays a secondary role in quantifying public sentiment towards the police.

City	TE	p-values
Atlanta	0.0315	<0.0001

Baltimore	0.0260	<0.0001
Boston	0.0352	<0.0001
Chicago	0.0327	<0.0001
Dallas	0.0327	<0.0001
Denver	0.0454	<0.0001
Detroit	0.0366	<0.0001
Houston	0.0228	<0.0001
Los Angeles	0.0334	<0.0001
Miami	0.0342	<0.0001
Minneapolis	0.0315	<0.0001
New York	0.0301	<0.0001
Philadelphia	0.0326	<0.0001
Phoenix	0.0415	<0.0001
Riverside	0.0236	<0.0001
San Francisco	0.0322	<0.0001
Tampa	0.0410	<0.0001
Washington	0.0333	<0.0001

4th Jan 24

Dear Mau,

Your manuscript titled "Understanding the role of media in the formation of public sentiment towards the police" has now been seen by our reviewers, whose comments appear below. In light of their advice I am delighted to say that we are happy, in principle, to publish a suitably revised version in Communications Psychology under the open access CC BY license (Creative Commons Attribution v4.0 International License).

We therefore invite you to revise your paper one last time to address a list of editorial requests. At the same time we ask that you edit your manuscript to comply with our format requirements and to maximise the accessibility and therefore the impact of your work.

EDITORIAL REQUESTS:

A key remaining issue is that the manuscript mentions the tool used to scrape Twitter (as it was then), but not whether data scraping used the dedicated API and complied with the ToS in place at the time of data access. Please provide us with detailed information on this matter and include mention details in the manuscript.

SUBMISSION INFORMATION:

OPEN ACCESS:

Communications Psychology is a fully open access journal. Articles are made freely accessible on publication under a [CC BY](http://creativecommons.org/licenses/by/4.0) license (Creative Commons Attribution 4.0 International License). This license allows maximum dissemination and re-use of open access materials and is preferred by many research funding bodies.

For further information about article processing charges, open access funding, and advice and support from Nature Research, please visit <https://www.nature.com/commspsychol/article-processing-charges>

At acceptance, you will be provided with instructions for completing this CC BY license on behalf of all authors. This grants us the necessary permissions to publish your paper. Additionally, you will be asked to declare that all required third party permissions have been obtained, and to provide billing information in order to pay the article-processing charge (APC).

* TRANSPARENT PEER REVIEW: Communications Psychology uses a transparent peer review system. On author request, confidential information and data can be removed from the published reviewer reports and rebuttal letters prior to publication. If you are concerned about the release of confidential data, please let us know specifically what information you would like to have removed. Please note that we cannot incorporate redactions for any other reasons.

* CODE AVAILABILITY: All Communications Psychology manuscripts must include a section titled "Code Availability" at the end of the methods section. We require that the custom analysis code supporting your conclusions is made available in a publicly accessible repository at this stage; please choose a repository that generates a digital object identifier (DOI) for the code; the link to the repository and the DOI must be included in the Code Availability statement. Publication as Supplementary Information will not suffice.

* DATA AVAILABILITY:

[link redacted]

Best regards,

Marike

Marike Schiffer, PhD
Chief Editor
Communications Psychology

REVIEWERS' COMMENTS:

Reviewer #4 (Remarks to the Author):

I am satisfied with the authors' response and the current version of the paper and recommend that it be published with no further revision.

Reviewer #6 (Remarks to the Author):

Thank you for making the suggested analyses and corrections.